# Provably Efficient $Q$-learning with Function Approximation via Distribution Shift Error Checking Oracle

**Simon S. Du**
Institute for Advanced Study
ssdu@ias.edu

**Yuping Luo**
Princeton University
yupingl@cs.princeton.edu

**Ruosong Wang**
Carnegie Mellon University
ruosongw@andrew.cmu.edu

**Hanrui Zhang**
Duke University
hrzhang@cs.duke.edu

## Abstract

$Q$-learning with function approximation is one of the most popular methods in reinforcement learning. Though the idea of using function approximation was proposed at least 60 years ago [27], even in the simplest setup, i.e, approximating $Q$-functions with linear functions, it is still an open problem how to design a provably efficient algorithm that learns a near-optimal policy. The key challenges are how to efficiently explore the state space and how to decide when to stop exploring *in conjunction with* the function approximation scheme.

The current paper presents a provably efficient algorithm for $Q$-learning with linear function approximation. Under certain regularity assumptions, our algorithm, Difference Maximization $Q$-learning (DMQ), combined with linear function approximation, returns a near-optimal policy using polynomial number of trajectories. Our algorithm introduces a new notion, the Distribution Shift Error Checking (DSEC) oracle. This oracle tests whether there exists a function in the function class that predicts well on a distribution $\mathcal{D}_1$, but predicts poorly on another distribution $\mathcal{D}_2$, where $\mathcal{D}_1$ and $\mathcal{D}_2$ are distributions over states induced by two different exploration policies. For the linear function class, this oracle is equivalent to solving a top eigenvalue problem. We believe our algorithmic insights, especially the DSEC oracle, are also useful in designing and analyzing reinforcement learning algorithms with general function approximation.

## 1 Introduction

$Q$-learning is a foundational method in reinforcement learning [35] and has been successfully applied in various domains. $Q$-learning aims at learning the optimal state-action value function ($Q$-function). Once we have learned the $Q$-function, at every state, we can just greedily choose the action with the largest $Q$ value, which is guaranteed to be an optimal policy.

Although being a fundamental method, theoretically, we only have a good understanding of $Q$-learning in the tabular setting. Strehl et al. [30] and Jin et al. [18] showed with proper exploration techniques, one can obtain a near-optimal $Q$-function (and so a near-optimal policy) using polynomial number of trajectories, in terms of number of states, actions and planning horizon. While these analyses provide valuable insights, they are of limited practical importance because the number of states in most applications is enormous. Even worse, it has been proved that in the tabular setting, the

number of trajectories needed to learn a near-optimal policy scales at least linearly with the number of states [16].

To resolve this problem, we need reinforcement learning methods that generalize, which, for $Q$-learning methods, is to constrain $Q$-function to a pre-specified function class, e.g., linear functions or neural networks. The basic assumption of this function approximation scheme is that the true $Q$-function lies in the function class. A natural problem is:

*Can we design provably efficient Q-learning algorithms with function approximation?*

Indeed, this is one of the major open problems in reinforcement learning [32]. The idea of using function approximation was proposed at least 60 years ago [27], where linear functions are used to approximate the value functions in playing checkers. However, even in the most basic setting, $Q$-learning with linear function approximation, there is no provably efficient algorithm in the general stochastic setting.

The key challenges are how to 1) efficiently explore the state space to learn a good predictor that generalizes across states and 2) decide when to stop exploring. In order to deal with these challenges, we need to exploit the fact that the true $Q$-function belongs to a pre-specified function class.

**Our Contributions**   Our main theoretical contribution is a provably efficient algorithm for $Q$-learning with linear function approximation in the episodic Markov decision process (MDP) setting.

**Theorem 1.1** (Main Theorem (informal))**.** *Suppose the Q-function is linear. Then under certain regularity assumptions, Algorithm 1, Difference Maximization $Q$-learning (DMQ) returns an $\epsilon$-suboptimal policy $\pi$ using $\mathrm{poly}(1/\epsilon)$ number of trajectories.*

Our algorithm works for episodic MDPs with general stochastic transitions. In contrast, previous algorithms only work for deterministic systems, or rely on strong assumptions, e.g., a sufficiently good exploration policy is given. See Section 2 for more discussion. Our main assumption is that the $Q$-function is linear. Note this is somehow a necessary assumption because otherwise one should not use linear function approximation in the first place.

Before getting into details, we first give an overview of our main techniques. As we have discussed, the main technical challenge is to design an efficient exploration algorithm, and decide when to stop exploring. Our main algorithmic contribution is to introduce a new notion, the Distribution Shift Error Checking (DSEC) oracle (cf. Oracle 4.1 and Oracle 4.2). Given two distributions $\mathcal{D}_1$ and $\mathcal{D}_2$, this oracle returns True if there exists a function in the pre-specified function class which predicts well on $\mathcal{D}_1$ but predicts poorly on $\mathcal{D}_2$. We will show that this is an extremely useful notion. If the oracle returns False, then our learned predictor performs well on both distributions. If the oracle returns True, we know $\mathcal{D}_2$ contains information that we can explore, which implies the policy that generates $\mathcal{D}_2$ is a valuable exploration policy. We will discuss the DSEC oracle in more detail in Section 4.

With this oracle at hand, a natural question is how many times this oracle will return True, as we will not stop exploring if it always returns True. A technical contribution of this paper is to show for the linear function class, this oracle will only return True at most polynomial number of times. At a high level, whenever the oracle returns True, it means we will learn something new from $\mathcal{D}_2$. However, since the complexity of the function class is bounded, we cannot learn new things too many times. Formally, we use a potential function argument to make this intuition rigorous (cf. Lemma A.5).

## 1.1   Organization

This paper is organized as follows. In Section 2, we review related work. In Section 3, we introduce necessary notations, definitions and assumptions. In Section 4, we describe the DSEC oracle in detail. In Section 5, we present our general algorithm for $Q$-learning with function approximation. In Section 6, we instantiate the general algorithm to the linear function approximation case, and present our main theorem. We conclude and discuss future works in Section 7. All technical proofs are deferred to the supplementary material.

## 2 Related Work

Classical theoretical reinforcement learning literature studies asymptotic behavior of concrete algorithms. The most related work is [24], which studies an online $Q$-learning algorithm with a fixed exploration policy. They showed that the estimated $Q$-function converges to the true $Q$-function asymptotically. Recently, Zou et al. [39] derived finite sample bounds for the same setting. The major drawback of these works is that they put strong assumptions on the fixed exploration policy. For example, Zou et al. [39] require that the covariance matrix induced by the exploration policy has lower bounded least eigenvalue. In general, it is hard to verify whether a policy has such benign properties.

While it is challenging to design efficient algorithms for $Q$-learning with function approximation, in the tabular setting, exploration becomes much easier, as one can first estimate the transition probabilities and then design exploration policies accordingly. There is a substantial body of work on tabular reinforcement learning [2, 16, 19, 5, 21, 10]. For $Q$-learning, Strehl et al. [30] introduced the delayed $Q$-learning algorithm which has $O(T^{4/5})$ regret bound. A recent work by Jin et al. [18] gave a UCB-based algorithm which enjoys $O\left(\sqrt{T}\right)$ regret bound. More recent papers provided refined analyses that exploit benign properties of the MDP, e.g., the gap between the optimal action and the rest [28, 38], which our algorithm also utilizes. However, it is hard to generalize the exploration techniques in these previous works, since they all rely on the fact that the total number of states is finite.

Recently, exploration algorithms are proposed for $Q$-learning with function approximation. Osband et al. [25] proposed a Thompson-sampling based method for the linear function class. Later works further generalized sampling-based algorithms to $Q$-functions with neural network parameterization [6, 23, 13]. However, none of these works have polynomial sample complexity guarantees. Pazis and Parr [26] gave a nearest-neighbor-based algorithm for exploration in continuous state space. However, in general this type of algorithms has exponential dependence on the state dimension.

The seminal work by Wen and Van Roy [36] proposed an algorithm, optimistic constraint propagation (OCP), which enjoys polynomial sample complexity bounds for a family of $Q$-function classes, including the linear function class as a special case. However, their algorithm can only deal with deterministic systems, i.e., both transition dynamics and rewards are deterministic. A line of recent papers study $Q$-learning in the general state-action metric space [37, 29]. However, due to the generality, the sample complexity has exponential dependence on the dimension.

Finally, a recent series of work introduced *contextual decision processes* (CDPs) [22, 17, 9, 31, 11] and developed algorithms with polynomial sample complexity guarantees. Our paper is not directly comparable with these results, since they can deal with general function classes. In some cases, the function approximation is even not for the $Q$-function, but for modeling the map from the observed state to hidden states [11]. The result in [17] also applies to our setting. However, their bound depends on both the function class complexity and a quantity called the Bellman rank. Conceptually, since our bound does not depend on the Bellman rank, our result thus demonstrates that the function class complexity alone is enough for efficient learning.

## 3 Preliminaries

**Notations** We begin by introducing necessary notations. We write $[h]$ to denote the set $\{1, \ldots, h\}$. For any finite set $S$, we write $\mathrm{unif}\,(S)$ to denote the uniform distribution over $S$ and $\triangle\,(S)$ to denote the probability simplex. Let $\|\cdot\|_2$ denote the Euclidean norm of a finite-dimensional vector in $\mathbb{R}^d$. For a symmetric matrix $A$, let $\|A\|_{\mathrm{op}}$ denote its operator norm and $\lambda_i\,(A)$ denote its $i$-th eigenvalue. Throughout the paper, all sets are multisets, i.e., a single element can appear multiple times.

**Markov Decision Processes (MDPs)** Let $\mathcal{M} = (\mathcal{S}, \mathcal{A}, H, P, R)$ be an MDP, where $\mathcal{S}$ is the (possibly uncountable) state space, $\mathcal{A}$ is the finite action space with $|\mathcal{A}| = K$, $H \in \mathbb{Z}_+$ is the planning horizon, $P : \mathcal{S} \times \mathcal{A} \to \triangle\,(\mathcal{S})$ is the transition function and $R : \mathcal{S} \times \mathcal{A} \to \triangle(\mathbb{R})$ is the reward distribution.

A (stochastic) policy $\pi : \mathcal{S} \to \triangle(\mathcal{A})$ prescribes a distribution over actions for each state. Without loss of generality, we assume a fixed start state $s_1$.[1] The policy $\pi$ induces a random trajectory $s_1, a_1, r_1, s_2, a_2, r_2, \ldots, s_H, a_H, r_H$ where $r_1 \sim R(s_1, a_1)$, $s_2 \sim P(s_1, a_1)$, $a_2 \sim \pi(s_2)$, etc. For a given policy $\pi$, we use $\mathcal{D}_h^\pi$ to denote the distribution over $\mathcal{S}_h$ induced by executing policy $\pi$.

To streamline our analysis, we denote $\mathcal{S}_h \subseteq \mathcal{S}$ to be the set of states at level $h$. Similar to previous theoretical reinforcement learning results, we also assume $r_h \geq 0$ for all $h \in [H]$ and $\sum_{h=1}^H r_h \leq 1$ [17]. Our goal is to find a policy $\pi$ that maximizes the expected reward $\mathbb{E}\left[\sum_{h=1}^H r_h \mid \pi\right]$. We use $\pi^*$ to denote the optimal policy.

Given a policy $\pi$, a level $h \in [H]$ and a state-action pair $(s, a) \in \mathcal{S}_h \times \mathcal{A}$, the $Q$-function is defined as $Q^\pi(s, a) = \mathbb{E}\left[\sum_{h'=h}^H r_{h'} \mid s_h = s, a_h = a, \pi\right]$. It will also be useful to define the value function of a given state $s \in \mathcal{S}_h$ as $V^\pi(s) = \mathbb{E}\left[\sum_{h'=h}^H r_{h'} \mid s_h = s, \pi\right]$. For simplicity, we denote $Q^*(s, a) = Q^{\pi^*}(s, a)$ and $V^* = V^{\pi^*}(s)$. Recall that if we know $Q^*$, we can just choose the action greedily: $\pi^*(s) = \operatorname{argmax}_{a \in \mathcal{A}} Q^*(s, a)$. In this paper, we make the following assumption about the variation of the suboptimality of policies [12].

**Assumption 3.1** (Bounded Coefficient of Variation of Policy Sub-optimality). *There exists a constant $1 \leq C < \infty$, such that for any fixed level $h \in [H]$ and deterministic policy $\pi$,*

$$\mathbb{E}_{s \sim \mathcal{D}_h^\pi}\left[|V^\pi(s) - V^*(s)|^2\right] \leq C \left(\mathbb{E}_{s \sim \mathcal{D}_h^\pi}\left[|V^\pi(s) - V^*(s)|\right]\right)^2.$$

Intuitively, this assumption says the variation due to the randomness over states is not too large comparing to the mean. For example, if the transition is deterministic, then this assumption holds with $C = 1$.

Our paper also relies on the following fine-grained characterization of the MDP.

**Definition 3.1** (Suboptimality Gaps). *Given $s \in \mathcal{S}$ and $a \in \mathcal{A}$, the gap is defined as $\operatorname{gap}(s, a) = V^*(s) - Q^*(s, a)$. The minimum gap is defined as $\gamma \triangleq \min_{s \in \mathcal{S}, a \in \mathcal{A}} \{\operatorname{gap}(s, a) : \operatorname{gap}(s, a) > 0\}$.*

This notion has been extensively studied in the bandit literature to obtain fine-grained bounds [4]. Recently, Simchowitz et al. [28] derived regret bounds in tabular MDPs based on this notion. In this paper we assume $\gamma > 0$, and the sample complexity of our algorithm depends polynomially on $1/\gamma$. Notice that assuming $\gamma$ is strictly positive is not a restrictive assumption for the *finite action setting* considered in this paper. First, in the contextual linear bandit literature, this assumption is widely discussed. See, e.g., [1, 8]. The notion, context, in the bandit literature is essentially $\phi(s)$ in our paper and the number of contexts can also be infinite. Second, there are many natural environments in RL which satisfy this assumption. For example, in many environments, states can be classified as good states and bad states. In these environments, an agent can obtain a reward only if it is in a good state. There are also two kinds of actions: good actions and bad actions. If the agent is in a good state and chooses a good action, the agent will transit to a good state. If the agent chooses a bad action, the agent will transit to a bad state. If the agent is in a bad state, whatever action the agent chooses, the agent will transit to a bad state. Note that for this kind of environments, there is a strictly positive gap between good actions and bad actions when the agent is in good states and there is no difference between good actions and bad actions when the agent is in bad states. In this case, $\gamma$ is strictly positive, since by Definition 3.1, we take the minimum over all state-action pairs with *strictly* positive gap. These environments are natural generalizations of the combination lock environment [20]. Some Atari games, e.g. Freeway, have a similar flavor as these environments.

**Function Approximation** When the state space is large, we need structures on the state space so that reinforcement learning methods can generalize. We constrain the optimal $Q$-function to a pre-specified function class $\mathcal{Q}$ [7], e.g., the class of linear functions. In this paper we associate each $h \in [H]$ and $a \in \mathcal{A}$ with a $Q$-function $f_h^a \in \mathcal{Q}$. We make the following assumption.

**Assumption 3.2.** *For every $(h, a) \in [H] \times \mathcal{A}$, its associated optimal $Q$-function is in $\mathcal{Q}$.*

This is a widely used assumption in the theoretical reinforcement learning literature [17]. Note that without this assumption, we cannot hope to obtain optimal policy using functions in $\mathcal{Q}$ as the $Q$-function.

The focus of this paper is about linear function class which is one of the most popular function classes used in practice. This function class depends on a feature extractor $\phi : \mathcal{S} \to \mathbb{R}^d$ which can be a hand-crafted feature extractor or a pre-trained neural network that transforms a state to a $d$-dimension embedding. For $s_h \in \mathcal{S}_h$ and $a \in \mathcal{A}$, our estimated optimal $Q$-function admits the form $f_h^a(s) = \phi(s)^\top \hat{\theta}_h^a$ where $\hat{\theta}_h^a \in \mathbb{R}^d$ only depends on the level $h \in [H]$ and $a \in \mathcal{A}$. Therefore, we only need to learn $K \cdot H$ $d$-dimension vectors (linear coefficients), since by Assumption 3.2, for each $h \in [H]$ and $a \in \mathcal{A}$, there exists $\theta_h^a \in \mathbb{R}^d$ such that for all $s_h \in \mathcal{S}_h$, $Q^*(s_h, a) = \phi(s_h)^\top \theta_h^a$.

The aim of this paper is to obtain polynomial sample complexity bounds. To this end, we also need some regularity conditions.

**Assumption 3.3.** *For all $s \in \mathcal{S}$, its feature is bounded $\|\phi(s)\|_2 \leq 1$. For all $h \in [H]$, $a \in \mathcal{A}$, the true linear predictor is bounded $\|\theta_h^a\|_2 \leq 1$.*

## 4 Distribution Shift Error Checking Oracle

As we have discussed in Section 1, in reinforcement learning, we often need to know whether a predictor learned from samples generated from one distribution $\mathcal{D}_1$ can predict well on another distribution $\mathcal{D}_2$. This is related to off-policy learning for which one often needs to bound the probability density ratio between $\mathcal{D}_1$ and $\mathcal{D}_2$ on all state-action pair. When function approximation scheme is used, we naturally arrive at the following oracle.

**Oracle 4.1** (Distribution Shift Error Checking Oracle $(\mathcal{D}_1, \mathcal{D}_2, \epsilon_1, \epsilon_2, \Lambda)$)**.** *For two given distributions $\mathcal{D}_1, \mathcal{D}_2$ over $\mathcal{S}$, two real numbers $\epsilon_1$ and $\epsilon_2$, and a regularizer $\Lambda : \mathcal{Q} \times \mathcal{Q} \to \mathbb{R}$, define*

$$v = \max_{f_1, f_2 \in \mathcal{Q}} \mathbb{E}_{s \sim \mathcal{D}_2} \left[ (f_1(s) - f_2(s))^2 \right]$$

$$s.t. \ \mathbb{E}_{s \sim \mathcal{D}_1} \left[ (f_1(s) - f_2(s))^2 \right] + \Lambda(f_1, f_2) \leq \epsilon_1.$$

*The oracle returns* True *if $v \geq \epsilon_2$, and* False *otherwise.*

To motivate this oracle, let $f_2$ be the optimal $Q$-function and $f_1$ is a predictor we learned using samples generated from distribution $\mathcal{D}_1$. In this scenario, we know $f_1$ has a small expected error $\epsilon_1$ on distribution $\mathcal{D}_1$. Note since we maximize over the entire function class $\mathcal{Q}$, $v$ is an upper bound on the expected error of $f_1$ on distribution $\mathcal{D}_2$. If $v$ is large enough, say larger than $\epsilon_2$, then it could be the case that we cannot predict well on distribution $\mathcal{D}_2$. On the other hand, if $v$ is small, we are certain that $f_1$ has small error on $\mathcal{D}_2$. Here we add a regularization term $\Lambda(f_1, f_2)$ to prevent pathological cases. The concrete choice of $\Lambda$ will be given later.

In practice, it is impossible to get access to the underlying distributions $\mathcal{D}_1$ and $\mathcal{D}_2$. Thus, we use samples generated from these two distributions instead.

**Oracle 4.2** (Sample-based Distribution Shift Error Checking Oracle $(D_1, D_2, \epsilon_1, \epsilon_2, \Lambda)$)**.** *For two set of states $D_1, D_2 \subseteq \mathcal{S}$, two real numbers $\epsilon_1$ and $\epsilon_2$, and a regularizer $\Lambda : \mathcal{Q} \times \mathcal{Q} \to \mathbb{R}$, define*

$$v = \max_{f_1, f_2 \in \mathcal{Q}} \frac{1}{|D_2|} \sum_{t_i \in D_2} \left[ (f_1(t_i) - f_2(t_i))^2 \right]$$

$$s.t. \ \frac{1}{|D_1|} \sum_{s_i \in D_1} \left[ (f_1(s_i) - f_2(s_i))^2 \right] + \Lambda(f_1, f_2) \leq \epsilon_1.$$

*The oracle returns* True *if $v \geq \epsilon_2$ and* False *otherwise. If $D_1 = \emptyset$, the oracle simply returns* True.

An interesting property of Oracle 4.2 is that it only depends on the states and does not rely on the reward values.

## 5 Difference Maximization $Q$-learning

Now we describe our algorithm. We maintain three sets of global variables.

---

**Algorithm 1** Difference Maximization $Q$-learning (DMQ)

---
**Output**: A near-optimal policy $\pi$.
1: **for** $h = H, H - 1, \ldots, 1$ **do**
2:     Run Algorithm 2 on input $h$.
3: Return $\hat{\pi}$, the greedy policy with respect to $\{f_h^a\}_{a \in \mathcal{A}, h \in [H]}$.

---

1. $\{f_h^a\}_{a \in \mathcal{A}, h \in [H]}$. These are our estimated $Q$-functions for all actions $a \in \mathcal{A}$ and all levels $h \in [H]$.
2. $\{\Pi_h\}_{h \in [H]}$. For each level $h \in [H]$, $\Pi_h$ is a set of exploration policies for level $h$, which we use to collect data.
3. $\{D_h\}_{h \in [H]}$. For each $h \in [H]$, $D_h = \{s_{h,i}\}_{i=1}^N$ is a set of states in $\mathcal{S}_h$.

We initialize these global variables in the following manner. For $\{f_h^a\}_{a \in \mathcal{A}, h \in [H]}$, we initialize them arbitrarily. For each $h \in [H]$, we initialize $\Pi_h$ to be a single purely random exploration policy, i.e., $\Pi_h = \{\pi\}$, where $\pi(s) = \text{unif}(\mathcal{A})$ for all $s \in \mathcal{S}$. We initialize $\{D_h\}_{h \in [H]}$ to be empty sets.

Algorithm 1 uses Algorithm 2 to learn predictors for each level $h \in [H]$. Algorithm 2 takes $h \in [H]$ as input, tries to learn predictors $\{f_h^a\}_{a \in \mathcal{A}}$ at level $h$. Algorithm 3 takes $h \in [H]$ and $a \in \mathcal{A}$ as inputs, and checks whether the predictors learned for later levels $h' > h$ are accurate enough under the current policy.

Now we explain Algorithm 2 and Algorithm 3 in more detail. Algorithm 2 iterates all actions, and for each action $a \in \mathcal{A}$, it uses Algorithm 3 to check whether we can learn the $Q$-function that corresponds to $a$ well. After executing Algorithm 3, we are certain that we can learn $f_h^a$ well (we will explain this in the next paragraph), and thus construct a set of new policies $\Pi_h^a = \{\pi_h^a\}_{\pi_h \in \Pi_h}$, in the following way. For each policy $\pi_h \in \Pi_h$, we define $\pi_h^a$ as

$$\pi_h^a(s_{h'}) = \begin{cases} \pi_h(s_{h'}) & \text{if } h' < h \\ a & \text{if } h' = h \\ \text{argmax}_{a' \in \mathcal{A}} f_{h'}^{a'}(s_{h'}) & \text{if } h' > h \end{cases} \quad (1)$$

This policy uses $\pi_h$ as the roll-in policy till level $h$, chooses action $a$ at level $h$ and uses greedy policy with respect to $\{f_{h'}^a\}_{h' > h, a \in \mathcal{A}}$, the current estimates of $Q$-functions at level $h + 1, \ldots, H$ as the roll-out policy. In each iteration, we sample one policy $\pi$ uniformly at random from $\Pi_h^a$, and use it to collect $(s, y)$, where $s \in \mathcal{S}_h$ and $y \in \mathbb{R}$ is the on-the-go reward. In total we collect a dataset $D_h^a$ with size $N \cdot |\Pi_h|$, and we use regression to learn a predictor on these data. Formally, we calculate

$$f_h^a = \text{argmin}_{f \in \mathcal{Q}} \left[ \frac{1}{N \cdot |\Pi_h|} \sum_{(s,y) \in D_h^a} (f(s) - y)^2 + \Gamma(f) \right]. \quad (2)$$

Here, $\Gamma(f)$ represents a regularization term on $f$. Finally, we update $D_h$ by using each $\pi_h \in \Pi_h$ to collect $N$ states in $\mathcal{S}_h$.

Now we explain Algorithm 3. For each $\pi_h \in \Pi_h$, we use $\pi_h^a$ defined in (1) to collect $N$ trajectories. For each $h' = h + 1, \ldots, H$, we set $\widetilde{D}_{\pi_h^a, h'} = \{s_{h',i}\}_{i=1}^N$, where $s_{h',i}$ is the state at level $h'$ in the $i$-th trajectory. Next, for each $h' = H, \ldots, h + 1$, we invoke Oracle 4.2 on input $D_{h'}$ and $\widetilde{D}_{\pi_h^a, h'}$. Note that $D_{h'}$ was collected when we execute Algorithm 2 to learn the predictors at level $h'$. The oracle will return whether our current predictors at level $h'$ can still predict well on the distribution that generates $\widetilde{D}_{\pi_h^a, h'}$. If not, then we add $\pi_h^a$ to our policy set $\Pi_{h'}$, and we execute Algorithm 2 to learn the predictors at level $h'$ once again. Note it is crucial to iterate $h'$ from $H$ to $h + 1$, so that we will always make sure the predictors at later levels are correct.

## 6 Provably Efficient $Q$-learning with Linear Function Approximation

Now we instantiate our algorithm to the linear function class. For the regression problem in (2), we set $\Gamma(\theta) = \lambda_{\text{ridge}} \|\theta\|_2^2$. The concrete choice of the parameter $\lambda_{\text{ridge}}$ will be given later. In this case,

## Algorithm 2

**Input**: $h \in [H]$, a target level.

1: **for** $a \in \mathcal{A}$ **do**
2:      Execute Algorithm 3 on input $(h, a)$.
3:      Initialize $D_h^a = \emptyset$.
4:      Construct a policy set $\Pi_h^a$ according to (1).
5:      **for** $i = 1, \ldots, N \cdot |\Pi_h^a|$ **do**
6:          Sample $\pi \sim \mathrm{unif}\,(\Pi_h^a)$.
7:          Use $\pi$ to collect $(s_i, y_i)$, where $s_i \in \mathcal{S}_h$ and $y_i$ is the on-the-go reward.
8:          Add $(s_i, y_i)$ into $D_h^a$.
9:      Learn a predictor $f_h^a = \mathrm{argmin}_{f \in \mathcal{Q}} \left[ \frac{1}{N \cdot |\Pi_h|} \sum_{(s,y) \in D_h^a} (f(s) - y)^2 + \Gamma(f) \right]$.
10: Set $D_h = \emptyset$.
11: **for** $\pi_h \in \Pi_h$ **do**
12:      Use $\pi_h$ to collect a set of states $\{s_{\pi_h, i}\}_{i=1}^N$, where $s_{\pi_h, i} \in \mathcal{S}_h$.
13:      Add all states $\{s_{\pi_h, i}\}_{i=1}^N$ into $D_h$.

## Algorithm 3

**Input**: target level $h \in [H]$ and an action $a \in \mathcal{A}$.

1: **for** $\pi_h \in \Pi_h$ **do**
2:      Collect $N$ trajectories using policy $\pi_h^a$ defined in (1).
3:      **for** $h' = H, H-1, \ldots, h+1$ **do**
4:          Let $\widetilde{D}_{\pi_h^a, h'} = \{s_{h', i}\}_{i=1}^N$ be the states at level $h'$ on the $N$ trajectories collected using $\pi_h^a$.
5:          Invoke Oracle 4.2 on input $\left( D_{h'}, \widetilde{D}_{\pi_h^a, h'}, \frac{\epsilon_s}{|\Pi_{h'}|}, \epsilon_t, \Lambda_{\Pi_{h'}} \right)$.
6:          **if** Oracle 4.2 returns True **then**
7:              $\Pi_{h'} = \Pi_{h'} \cup \{\pi_h^a\}$.
8:              Execute Algorithm 2 on input $h'$.

the regression program represents the ridge regression estimator

$$\hat{\theta}_h^a = \left( \frac{1}{|D_h^a|} \sum_{(s,y) \in D_h^a} \phi(s)\phi(s)^\top + \lambda_{\mathrm{ridge}} \cdot I \right)^{-1} \left( \frac{1}{|D_h^a|} \sum_{(s,y) \in D_h^a} y \cdot \phi(s) \right),$$

and $f_h^a(s_h) = \phi(s_h)^\top \hat{\theta}_h^a$ for $s_h \in \mathcal{S}_h$.

For Oracle 4.2, we choose $\Lambda_{\Pi_{h'}}(\theta_1, \theta_2) = \lambda_r / |\Pi_{h'}| \cdot \|\theta_1 - \theta_2\|_2^2$. The concrete choice of the parameter $\lambda_r$ will be given later. Since $\mathcal{Q}$ is the linear function class, the optimization problem is equivalent to the following program

$$\max_{\theta_1, \theta_2} \frac{1}{|D_2|} \sum_{t_i \in D_2} \left( (\theta_1 - \theta_2)^\top \phi(t_i) \right)^2$$

$$\text{s.t.} \quad \frac{1}{|D_1|} \sum_{s_i \in D_1} \left( (\theta_1 - \theta_2)^\top \phi(s_i) \right)^2 + \lambda_r / |\Pi_{h'}| \cdot \|\theta_1 - \theta_2\|_2^2 \le \epsilon_1.$$

We let $M_1 = \frac{1}{|D_1|} \sum_{s_i \in D_1} \phi(s_i)\phi(s_i)^\top + \lambda_r / |\Pi_{h'}| \cdot I$, $M_2 = \frac{1}{|D_2|} \sum_{t_i \in D_2} \phi(t_i)\phi(t_i)^\top$, and let $\tilde{\theta} \triangleq \frac{1}{\sqrt{\epsilon_1}} M_1^{1/2} (\theta_1 - \theta_2)$, then the optimization problem can be further reduced to

$$\max_{\tilde{\theta}} \tilde{\theta}^\top \left( \epsilon_1 M_1^{-\frac{1}{2}} M_2 M_1^{-\frac{1}{2}} \right) \tilde{\theta} \quad \text{s.t.} \quad \left\| \tilde{\theta} \right\|_2 \le 1,$$

which is equivalent to compute the top eigenvalue of $\epsilon_1 M_1^{-\frac{1}{2}} M_2 M_1^{-\frac{1}{2}}$. Therefore, the regression problem in (2) and Oracle 4.2 can be efficiently implemented. Our main result is the following theorem.

**Theorem 6.1** (Provably Efficient $Q$-Learning with Linear Function Approximation). *Let $\epsilon \leq$ poly$(\gamma, 1/C, 1/d, 1/H, 1/K)$ be the target accuracy parameter. Under Assumption 3.1, 3.2 and 3.3, then using at most poly$(1/\epsilon)$ trajectories, with high probability, Algorithm 1 returns a policy $\hat{\pi}$ that satisfies $V^{\hat{\pi}}(s_1) \geq V^*(s_1) - \epsilon$.*

This theorem demonstrates that if the true $Q$-function is linear, then it is actually possible to learn a near-optimal policy with polynomial number of samples. We refer readers to the Proof of Theorem 6.1 for the specific values of $\epsilon_t, \epsilon_s, \epsilon_N, \lambda_{\mathrm{ridge}}, \lambda_r, N$. Furthermore, our algorithm also runs in polynomial time. Therefore, this is the first provably efficient algorithm for $Q$-learning with function approximation in the stochastic setting.

Now we briefly sketch the proof of Theorem 6.1. The full proof is deferred to Section A. Our proof follows directly from the design of our algorithm. First, through classical analysis of linear regression, we know the learned predictor $\hat{\theta}_h^a$ can predict well on the distribution induced by $\pi_h^a$. Second, Oracle 4.2 guarantees that if it returns False, then the learned predictors at level $h'$ can predict well on the distribution over $\mathcal{S}_h$ induced by the policy $\pi_h^a$. Therefore, the labels we used to learn $\theta_a^h$ have only small bias, and thus, we can learn $\theta_a^h$ well. Now the trickiest part of the proof is to show Oracle 4.2 returns True at most polynomial number of times. To establish this, for each $h \in [H]$, we construct a potential function in terms of covariance matrices induced by the policies in $\Pi_h$. We show whenever a new policy is added to $\Pi_h$, this potential function must be increased by a multiplicative factor. We further show this potential function is at always polynomially upper bounded by the size of the policy set. Therefore, we can conclude the size of $\Pi_h$ is polynomially upper bounded. See Lemma A.5 for details.

## 7 Discussion

By giving a provably efficient algorithm for $Q$-learning with function approximation, this paper paves the way for rigorous studies of modern model-free reinforcement learning methods with function approximation. Now we list some future directions.

**Regret Bound** This paper presents a PAC bound but no regret bound. Note that we assume the gap between the on-the-go reward of the best action and the rest is strictly positive. In the tabular setting, previous work showed that under this assumption, one can obtain $\log T$ regret bound [28, 38]. We believe it would be a very strong result to prove (or disprove) $\log T$ regret bound in the setting considered in this paper.

**$Q$-learning with General Function Class** While the main theorem in this paper is about the linear function class, the DSEC oracle and the general algorithmic framework applies to any function classes. From an information-theoretic point of view, given Oracle 4.2, can we use it to design algorithms for general function class with polynomial sample complexity guarantees? For example, if the $Q$-function class has a bounded VC-dimension, can Algorithm 1 give a polynomial sample complexity guarantee? We believe a generalization of Lemma A.5 is required to resolve this question. Another interesting problem is to generalize our algorithm to the case that the $Q$-function is not exactly linear but can only be approximated by a linear function.

From the computational point of view, can we characterize the function classes for which we have an efficient solver for Oracle 4.2? For those we do not have such exact solvers, can we develop a relaxed version of Oracle 4.2 which, possibly sacrificing the sample efficiency, makes the optimization problem tractable. This idea was used in the sparse learning literature [34]. Another interesting problem is to improve the computational efficiency of our algorithm to make it fast enough to be used in practice.

**Toward a Rigorous Theory for DQN** Deep $Q$-learning (DQN) is one of the most popular model-free methods in modern reinforcement learning. Recent studies established that over-parameterized neural networks are equivalent to kernel predictors [15, 3] with multi-layer kernel functions. Since kernel predictors can be viewed as linear predictors in infinite dimensional feature spaces, can we adapt our algorithm to over-parameterized neural networks and multi-layer kernels, and prove polynomial sample complexity guarantees when, e.g., the true $Q$-function has a small reproducing Hilbert space norm?

## Acknowledgements

The authors would like to thank Nan Jiang, Akshay Krishnamurthy, Wen Sun, Yining Wang and Lin F. Yang for useful discussions. The paper was initiated while S. S. Du was an intern at MSR NYC and a Ph.D. student at Carnegie Mellon University. Part of this work was done while S. S. Du and R. Wang were visiting Simons Institute.

## Footnotes

[1]Some papers assume the starting state is sampled from a distribution $P_1$. Note this is equivalent to assuming a fixed state $s_1$, by setting $P(s_1, a) = P_1$ for all $a \in \mathcal{A}$ and now our $s_2$ is equivalent to the starting state in their assumption.

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
