[Supplementary Material]

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

## A    Proof of Theorem 6.1

In this section we give the proof of Theorem 6.1.

*Proof of Theorem 6.1.*  In the proof we set $\epsilon_t = \frac{\gamma^2 \epsilon}{5H}$, $\epsilon_s = 96C \cdot \epsilon^2 d \log (d/\epsilon)$, $\epsilon_N = \epsilon^2$, $\lambda_{\text{ridge}} = \epsilon^2$, $\lambda_r = \epsilon^6$, $B = 12d \log(d/\epsilon)$ and $N = \frac{d}{\lambda_r^2} \text{polylog}(\frac{1}{\epsilon})$. First, by Lemma A.6, we know with high probability over the generating process of $\{D_h\}_{h \in [H]}$, we have $|\Pi_h| \leq B$ for all $h \in [H]$. Note this also shows our algorithm ends in polynomial time. In the following we condition on this event.

We will prove $\hat{\pi}$ satisfies

$$\mathbb{P}_{s_h \sim \mathcal{D}_h^{\hat{\pi}}} [\hat{\pi}(s_h) \neq \pi^*(s_h)] \leq \frac{\epsilon}{H}$$

for all $h \in [H]$. By Lemma A.1, we know this implies the main conclusion.

To prove this property, it suffices to show our estimated predictors satisfy

$$\mathbb{P}_{s_h \sim \mathcal{D}_h^{\hat{\pi}}} \left[ |f_h^a(s_h) - Q^*(s_h, a)| \leq \frac{\gamma}{2} \right] \leq \frac{\epsilon}{H}$$

for all $h \in [H]$ and $a \in \mathcal{A}$. Since $f_h^a(s_h) = \phi(s_h)^\top \hat{\theta}_h^a$, by Markov's inequality, we only need to show

$$\mathbb{E}_{s_h \sim \mathcal{D}_h^{\hat{\pi}}} \left[ \left( \left( \hat{\theta}_h^a \right)^\top \phi(s_h) - Q^*(s_h, a) \right)^2 \right] \leq \frac{\gamma^2 \epsilon}{4H}. \tag{3}$$

We prove the following two invariants regarding the algorithm.

1. Each time we update $f_h^a$ (and thus $\hat{\theta}_h^a$) in Line 9 of Algorithm 2, for all $\pi \in \Pi_h$, it holds that

$$\mathbb{E}_{s_h \sim \mathcal{D}_h^\pi} \left[ \left( \left( \hat{\theta}_h^a \right)^\top \phi(s_h) - Q^*(s_h, a) \right)^2 \right] \leq \frac{\gamma^2 \epsilon}{4H}$$

and $\left\| \hat{\theta}_h^a \right\|_2 \leq 1/\lambda_{\text{ridge}}$.

2. Each time Oracle 4.2 returns False in Line 5 of Algorithm 3, it holds that for all actions $a' \in \mathcal{A}$,

$$\mathbb{E}_{s_{h'} \sim \mathcal{D}_{h'}^{\pi_h^a}} \left[ \left( \left( \hat{\theta}_{h'}^{a'} \right)^\top \phi(s_{h'}) - Q^*(s_{h'}, a') \right)^2 \right] \leq \frac{\gamma^2 \epsilon}{4H}.$$

These two invariants imply that for the policy $\hat{\pi}$ returned by Algorithm 1, (3) holds for all $h \in [H]$ and $a \in \mathcal{A}$, which also implies the correctness of our algorithm. It remains to prove these two invariants.

We first prove the first invariant. We fix a level $h \in [H]$ and an action $a \in \mathcal{A}$. Let $f_h^a \in \mathcal{Q}$ be the predictor calculated in Line 9 of Algorithm 2, and $\hat{\theta}_h^a \in \mathbb{R}^d$ be the corresponding linear coefficients. Since $D_h^a$ is collected after executing Line 2 of Algorithm 2, by induction, for all policies $\pi_h^a \in \Pi_h^a$ defined in (1), $h' > h$ and $a' \in \mathcal{A}$, we have

$$\mathbb{E}_{s_{h'} \sim \mathcal{D}_{h'}^{\pi_h^a}} \left[ \left( \left( \hat{\theta}_{h'}^{a'} \right)^\top \phi(s_{h'}) - Q^*(s_{h'}, a') \right)^2 \right] \leq \frac{\gamma^2 \epsilon}{4H}.$$

It follows that for all policies $\pi_h^a \in \Pi_h^a$, $h' > h$ and $a' \in \mathcal{A}$, we have

$$\mathbb{P}_{s_{h'} \sim \mathcal{D}_{h'}^{\pi_h^a}} \left[ \left| \left( \hat{\theta}_{h'}^{a'} \right)^\top \phi(s_{h'}) - Q^*(s_{h'}, a') \right| \geq \gamma/2 \right] \leq \epsilon/H.$$

Thus, by Lemma A.1

$$\mathbb{E}_{s_{h+1}\sim\mathcal{D}^{\pi_h^a}_{h+1}}\left[V^{\pi_h^a}(s_{h+1})\right] \geq \mathbb{E}_{s_{h+1}\sim\mathcal{D}^{\pi_h^a}_{h+1}}\left[V^*(s_{h+1})\right] - \epsilon.$$

By Assumption 3.1, for all $\pi_h^a \in \Pi_h^a$, we have

$$\mathbb{E}_{s_{h+1}\sim\mathcal{D}^{\pi_h^a}_{h+1}}\left[\left(V^{\pi_h^a}(s_{h+1}) - V^*(s_{h+1})\right)^2\right] \leq C\varepsilon^2,$$

which implies

$$\mathbb{E}_{\pi_h^a\sim\mathrm{unif}\left(\Pi_h^a\right),s_{h+1}\sim\mathcal{D}^{\pi_h^a}_{h+1}}\left[\left(V^{\pi_h^a}(s_{h+1}) - V^*(s_{h+1})\right)^2\right] \leq C\varepsilon^2,$$

For each $(s_i, y_i) \in D_h^a$, we have

$$y_i = \phi(s_i)^\top \theta_h^a + b_i + \xi_i,$$

where $\mathbb{E}[b_i^2] \leq C\varepsilon^2$, $|\xi_i| \leq 1$ almost surely and $\mathbb{E}[\xi_i] = 0$.

By Lemma A.3,

$$\left(\hat{\theta}_h^a - \theta_h^a\right)^\top \mathbb{E}_{\pi_h\sim\mathrm{unif}(\Pi_h),s_h\sim\mathcal{D}^{\pi_h}_h}\left[\phi(s_h)\phi(s_h)^\top\right]\left(\hat{\theta}_h^a - \theta_h^a\right) \leq 4\left(C\epsilon^2 + \epsilon_N + \lambda_{\mathrm{ridge}}\right),$$

which implies for all $\pi_h \in \Pi_h$,

$$\left(\hat{\theta}_h^a - \theta_h^a\right)^\top \mathbb{E}_{s_h\sim\mathcal{D}^{\pi_h}_h}\left[\phi(s_h)\phi(s_h)^\top\right]\left(\hat{\theta}_h^a - \theta_h^a\right) \leq 4\left(C\epsilon^2 + \epsilon_N + \lambda_{\mathrm{ridge}}\right)\cdot B \leq \frac{\gamma^2\epsilon}{4H}.$$

By Lemma A.4, we have $\left\|\hat{\theta}_h^a\right\|_2 \leq 1/\lambda_{\mathrm{ridge}}$. Thus, the first invariant holds.

Now we can prove the second invariant. By the first invariant, using Lemma A.2 and Lemma A.4, with high probability, we have

$$\left(\hat{\theta}_{h'}^a - \theta_{h'}^a\right)^\top \left(\lambda_r I + \sum_{\pi_{h'}\in\Pi_{h'}}\sum_{i=1}^N \frac{1}{N}\left[\phi(s_{i,\pi_{h'}})\phi(s_{i,\pi_{h'}})^\top\right]\right)\left(\hat{\theta}_{h'}^a - \theta_{h'}^a\right)$$

$$\leq 5\left(C\epsilon^2 + \epsilon_N + \lambda_{\mathrm{ridge}} + \lambda_r/\lambda_{\mathrm{ridge}}^2\right)|\Pi_{h'}| \leq \epsilon_s.$$

Since Oracle 4.2 returns False, we know

$$\left(\hat{\theta}_{h'}^a - \theta_{h'}^a\right)^\top \left(\sum_{i=1}^N \frac{1}{N}\left[\phi\left(s_{h',i}\right)\phi\left(s_{h',i}\right)^\top\right]\right)\left(\hat{\theta}_{h'}^a - \theta_{h'}^a\right) \leq \epsilon_t.$$

By Lemma A.2,

$$\left(\hat{\theta}_h^a - \theta_h^a\right)^\top \left(\mathbb{E}_{s_{h'}\sim\mathcal{D}^{\pi_a}_{h'}}\left[\phi(s_{h'})\phi\left(s_{h'}\right)^\top\right]\right)\left(\hat{\theta}_h^a - \theta_h^a\right) \leq \epsilon_t + \lambda_r(1 + 1/\lambda_{\mathrm{ridge}}^2) \leq \frac{\gamma^2\epsilon}{4H}.$$

This finishes the proof. $\qquad\square$

**Lemma A.1** (Policy Correctness). *Given fixed level $h \in [H]$ and a policy $\pi$, suppose for all $h' = h, \ldots, H$,*

$$\mathbb{P}_{s_{h'}\sim\mathcal{D}^{\pi}_{h'}}\left[\pi(s_{h'}) \neq \pi^*(s_{h'})\right] \leq \epsilon/H.$$

*Then for all $h' = h, \ldots, H$,*

$$\mathbb{E}_{s_{h'}\sim\mathcal{D}^{\pi}_{h'}}\left[V^\pi(s_{h'})\right] \geq \mathbb{E}_{s_{h'}\sim\mathcal{D}^{\pi}_{h'}}\left[V^*(s_{h'})\right] - \epsilon.$$

*Proof of Lemma A.1.* Consider the random trajectory $s_1, a_1, r_1, s_2, a_2, r_2, \ldots, s_H, a_H, r_H$ induced by policy $\pi$. With probability at least $1 - \epsilon$, we have $\pi(s_{h'}) = \pi^*(s_{h'})$ for all $h' = h, \ldots, H$. We prove the claim by using $\sum_{h=1}^H r_h \leq 1$.

$\qquad\square$

**Lemma A.2** (Covariance Concentration Bound [33]). *Suppose $M_1, \ldots, M_N \in \mathbb{R}^{d \times d}$ are i.i.d. drawn from a distribution $\mathcal{D}$ over positive semi-definite matrices. If $\|M_t\|_F \leq 1$ almost surely and $N = \Omega\left(\frac{d \log(d/\delta)}{\epsilon^2}\right)$, then with probability at least $1 - \delta$,*

$$\left\| \frac{1}{N} \sum_{t=1}^{N} M_t - \mathbb{E}_{M \sim \mathcal{D}}[M] \right\|_{op} \leq \epsilon.$$

**Lemma A.3** (Ridge Regression with Bias). *Suppose*

$$(s_1, y_1), (s_2, y_2), \ldots, (s_N, y_N)$$

*are i.i.d. drawn from a distribution $\mathcal{D}$ and satisfy*

$$y_i = \theta^\top \phi(s_i) + b_i + \xi_i,$$

*where $\mathbb{E}_{(s_i, y_i) \sim \mathcal{D}}[b_i^2] \leq \bar{b}^2$ for some $0 \leq \bar{b} \leq 1$, $|\xi_i| \leq 1$ almost surely and $\mathbb{E}[\xi_i] = 0$. Let*

$$S = \left[\phi(s_1)^\top; \ldots; \phi(s_N)^\top\right] \in \mathbb{R}^{N \times d},$$

$$y = [y_1; \ldots; y_N] \in \mathbb{R}^N,$$

*and*

$$\hat{\theta} = \left(\frac{S^\top S}{N} + \lambda_{\mathrm{ridge}} \cdot I\right)^{-1} \frac{S^\top y}{N}$$

*be the ridge regression estimator. If $N = \Omega\left(\frac{d}{\epsilon_N^2} \log\left(\frac{d}{\delta}\right)\right)$, then with probability at least $1 - \delta$,*

$$\mathbb{E}_{s \sim \mathcal{D}}\left[\left(\left(\theta - \hat{\theta}\right)^\top \phi(s)\right)^2\right] \leq 4\left(\bar{b}^2 + \epsilon_N + \lambda_{\mathrm{ridge}}\right).$$

*Proof of Lemma A.3.* Let

$$b = [b_1; \ldots; b_N] \in \mathbb{R}^N.$$

By Chernoff bound, with probability at least $1 - \delta/3$, $\|b\|_2^2/N \leq \bar{b}^2 + \varepsilon_N/2$.

$$
\begin{aligned}
&\hat{\theta} - \theta \\
&= \left(\frac{S^\top S}{N} + \lambda_{\mathrm{ridge}} \cdot I\right)^{-1} \frac{S^\top y}{N} - \theta \\
&= \left(\frac{S^\top S}{N} + \lambda_{\mathrm{ridge}} \cdot I\right)^{-1} \frac{S^\top (S\theta + b + \xi)}{N} - \theta \\
&= \left(\left(\frac{S^\top S}{N} + \lambda_{\mathrm{ridge}} \cdot I\right)^{-1} \frac{S^\top S}{N} - I\right)\theta + \left(\frac{S^\top S}{N} + \lambda_{\mathrm{ridge}} \cdot I\right)^{-1} \frac{S^\top b}{N} + \left(\frac{S^\top S}{N} + \lambda_{\mathrm{ridge}} \cdot I\right)^{-1} \frac{S^\top \xi}{N} \\
&\triangleq I_1 + I_2 + I_3.
\end{aligned}
$$

Thus,

$$
\begin{aligned}
&\frac{1}{N} \left\| S\left(\hat{\theta} - \theta\right) \right\|_2^2 \\
&= \left(\hat{\theta} - \theta\right)^\top \frac{S^\top S}{N} \left(\hat{\theta} - \theta\right) \\
&= (I_1 + I_2 + I_3)^\top \frac{S^\top S}{N} (I_1 + I_2 + I_3) \\
&\leq 3\left(I_1^\top \frac{S^\top S}{N} I_1 + I_2^\top \frac{S^\top S}{N} I_2 + I_3^\top \frac{S^\top S}{N} I_3\right).
\end{aligned}
$$

For the first term, we have

$$I_1^\top \frac{S^\top S}{N} I_1 \leq \|\theta\|_2^2 \cdot \left\| \left( \left( \frac{S^\top S}{N} + \lambda_{\text{ridge}} \cdot I \right)^{-1} \frac{S^\top S}{N} - I \right)^\top \frac{S^\top S}{N} \left( \left( \frac{S^\top S}{N} + \lambda_{\text{ridge}} \cdot I \right)^{-1} \frac{S^\top S}{N} - I \right) \right\|_{\text{op}}$$

$$\leq \lambda_{\text{ridge}} \|\theta\|_2^2 \leq \lambda_{\text{ridge}}.$$

For the second term, we have

$$I_2^\top \frac{S^\top S}{N} I_2 \leq \frac{\|b\|_2^2}{N} \left\| \frac{1}{N} \left( \left( \frac{S^\top S}{N} + \lambda_{\text{ridge}} \cdot I \right)^{-1} S^\top \right)^\top \frac{S^\top S}{N} \left( \left( \frac{S^\top S}{N} + \lambda_{\text{ridge}} \cdot I \right)^{-1} S^\top \right) \right\|_{\text{op}} \leq \frac{\|b\|_2^2}{N}.$$

Using Lemma 29 of [14], we know $I_3^\top \frac{S^\top S}{N} I_3 \leq \frac{\epsilon_N}{2}$ with probability at least $1 - \delta/3$. Finally, applying the standard empirical process method, we can bound the generalization error and thus finish the proof.

$\square$

**Lemma A.4** (Ridge Regression Gives Upper Bound on the Norm). *Suppose*

$$(\phi(s_1), y_1), (\phi(s_2), y_2), \ldots, (\phi(s_N), y_N) \in \mathbb{R}^d \times \mathbb{R}$$

*satisfy* $|y_i| \leq 1$ *and* $\|\phi(s_i)\|_2 \leq 1$ *for all* $i \in [N]$*. Let* $S = \left[ \phi(s_1)^\top; \ldots; \phi(s_N)^\top \right] \in \mathbb{R}^{N \times d}$, $y = [y_1; \ldots; y_N] \in \mathbb{R}^N$ *and* $\hat\theta = \left( \frac{S^\top S}{N} + \lambda_{\text{ridge}} \cdot I \right)^{-1} \frac{S^\top y}{N}$ *be the ridge regression estimator. Then we have*

$$\left\| \hat\theta \right\|_2 \leq \frac{1}{\lambda_{\text{ridge}}}.$$

*Proof of Lemma A.4.* By triangle inequality, $\left\| \frac{S^\top y}{N} \right\|_2 \leq 1$. Furthermore,

$$\left\| \left( \frac{S^\top S}{N} + \lambda_{\text{ridge}} \cdot I \right)^{-1} \right\|_{\text{op}} \leq 1/\lambda_{\text{ridge}}.$$

Thus,

$$\left\| \hat\theta \right\|_2 \leq \frac{1}{\lambda_{\text{ridge}}}.$$

$\square$

**Lemma A.5.** *Consider the following process. Initially* $\mathcal{M} = \emptyset$*. Let* $\epsilon_s$ *and* $\epsilon_t$ *be two real numbers such that* $0 < \epsilon_s \leq \frac{\epsilon_t}{d}$*. For* $t = 1, 2, \ldots$*, we receive a positive semi-definite matrix* $M_t \in \mathbb{R}^{d \times d}$ *with* $\|M_t\|_F \leq 1$*. If there exists* $x \in \mathbb{R}^d$ *such that* $x^\top M_t x \geq \epsilon_t$ *and* $x^\top \left( \lambda_r I + \sum_{M \in \mathcal{M}} M \right) x \leq \epsilon_s$*, then we add* $M_t$ *into* $\mathcal{M}$*. It holds that* $|\mathcal{M}| \leq 2d \log \left( \frac{d}{\lambda_r} \right)$ *throughout the process.*

*Proof of Lemma A.5.* We first show that if $\mathcal{M} \neq \emptyset$, then after adding $M_t$ into $\mathcal{M}$, we must have

$$\det \left( \lambda_r I + \sum_{M \in \mathcal{M}} M \right) \geq \left( 1 + \frac{\epsilon_t}{d\epsilon_s} \right) \det \left( \lambda_r I + \sum_{M \in \mathcal{M} \setminus \{M_t\}} M \right) \geq 2 \det \left( \lambda_r I + \sum_{M \in \mathcal{M} \setminus \{M_t\}} M \right).$$

Let

$$A = \lambda_r I + \sum_{M \in \mathcal{M} \setminus \{M_t\}} M,$$

and

$$A = \sum_{j=1}^d \lambda_j v_j v_j^\top$$

be its spectral decomposition. For the positive semi-definite matrix $M_t$, we write $M_t = UU^\top$. By matrix determinant lemma, we have

$$\det\left(A + M_t\right) = \det\left(I + U^\top A^{-1} U\right) \det\left(A\right).$$

Therefore, it suffices to prove the largest eigenvalue of $U^\top A_i^{-1} U$ is larger than $\frac{\epsilon_t}{d\epsilon_t} \geq 1$.

Since we add $M_t$ into $\mathcal{M}$, there exists $x$ such that

$$\|Ux\|_2^2 \geq \epsilon_t$$

and

$$\sum_{j=1}^{d} \lambda_j \left(v_j^\top x\right)^2 \leq \epsilon_s.$$

Let $z = Ux/\|Ux\|$ be a unit vector such that $z^\top Ux = \|Ux\|_2$. Let $u = Uz$. We have

$$
\begin{aligned}
z^\top U^\top A_i^{-1} U z &= u^\top \left(\sum_{j=1}^{d} \frac{1}{\lambda_j} v_j v_j^\top\right) u \\
&= \sum_{j=1}^{d} \frac{1}{\lambda_j} \left(v_j^\top u\right)^2 \\
&= \sum_{j=1}^{d} \left(v_j^\top u\right)^2 \left(v_j^\top x\right)^2 \frac{1}{\lambda_j \cdot \left(v_j^\top x\right)^2} \\
&\geq \frac{1}{\epsilon_s} \sum_{j=1}^{d} \left(v_j^\top u\right)^2 \left(v_j^\top x\right)^2 \\
&\geq \frac{1}{\epsilon_s d} \left(\sum_{j=1}^{d} u^\top v_j v_j^\top x\right)^2 \\
&= \frac{1}{\epsilon_s d} \left(z^\top Ux\right)^2 \\
&= \frac{1}{\epsilon_s d} \|Ux\|_2^2 \\
&\geq \frac{\epsilon_t}{d\epsilon_s},
\end{aligned}
$$

which implies

$$\det\left(\lambda_r I + \sum_{M \in \mathcal{M}} M\right) \geq 2 \det\left(\lambda_r I + \sum_{M \in \mathcal{M} \setminus \{M_t\}} M\right).$$

It follows that

$$\det\left(\lambda_r I + \sum_{M \in \mathcal{M}} M\right) \geq 2^{|\mathcal{M}|-1} \lambda_r^d.$$

On the other hand, we have

$$\det\left(\lambda_r I + \sum_{M \in \mathcal{M}} M\right) \leq \left(|\mathcal{M}| + 1\right)^d,$$

since $\|M\|_F \leq 1$ for all $M \in \mathcal{M}$.

Thus,

$$2^{|\mathcal{M}|-1} \lambda_r^d \leq \left(|\mathcal{M}| + 1\right)^d,$$

which implies $|\mathcal{M}| \leq 2d \log\left(\frac{d}{\lambda_r}\right)$. $\qquad\square$

**Lemma A.6** (Polynomially Bounded Policy Set)**.** *If* $N \geq \frac{d}{\lambda_r^2} \cdot B^2 \cdot \text{polylog}\,(1/\epsilon)$, *then with high probability we have* $|\Pi_h| \leq B$ *for all* $h \in [H]$.

*Proof of Lemma A.6.* Consider a fixed $h' \in [H]$. We will add a new policy $\pi_h^a$ into $\Pi_{h'}$ only when Oracle 4.2 returns True in Line 6 of Algorithm 3. Since Oracle 4.2 returns True in Line 6 of Algorithm 3, there must exist $x = \theta_1 - \theta_2$ such that

$$x^\top \left( \lambda_r / |\Pi_{h'}| \cdot I + \frac{1}{N|\Pi_{h'}|} \sum_{\pi_{h'} \in \Pi_{h'}} \sum_{i=1}^{N} \phi(s_{\pi_{h'},i}) \phi(s_{\pi_{h'},i})^\top \right) x \leq \epsilon_s / |\Pi_{h'}|.$$

Recall that $D_{h'} = \{s_{\pi_{h'},i}\}_{\pi_{h'} \in \Pi_{h'}, i \in [N]}$ is a set of states generated using policies in $\Pi_{h'}$ (cf. Line 13 of Algorithm 2). It follows that

$$x^\top \left( \lambda_r \cdot I + \sum_{\pi_{h'} \in \Pi_{h'}} \frac{1}{N} \sum_{i=1}^{N} \phi(s_{\pi_{h'},i}) \phi(s_{\pi_{h'},i})^\top \right) x \leq \epsilon_s.$$

Thus, we have

$$\|x\|_2^2 \leq \epsilon_s / \lambda_r.$$

By Lemma A.2, for each $\pi_{h'} \in \Pi_{h'}$, with high probability,

$$\left| x^\top \left( \frac{1}{N} \sum_{i=1}^{N} \phi(s_{\pi_{h'},i}) \phi(s_{\pi_{h'},i})^\top - \mathbb{E}_{s_{\pi_{h'}} \sim \mathcal{D}_{h'}^{\pi_{h'}}} \left[ \phi(s_{\pi_{h'}}) \phi(s_{\pi_{h'}})^\top \right] \right) x \right| \leq \epsilon_s / B.$$

Therefore, with high probability, it is satisfied that

$$x^\top \left( \lambda_r I + \sum_{\pi_{h'} \in \Pi_{h'}} \mathbb{E}_{s_{\pi_{h'}} \sim \mathcal{D}_{h'}^{\pi_{h'}}} \left[ \phi(s_{\pi_{h'}}) \phi(s_{\pi_{h'}})^\top \right] \right) x \leq 2\epsilon_s.$$

Moreover, since Oracle 4.2 returns True in Line 6 of Algorithm 3, we must have

$$x^\top \left( \frac{1}{N} \sum_{i=1}^{N} \left[ \phi(s_{h',i}) \phi(s_{h',i})^\top \right] \right) x \geq \epsilon_t.$$

Recall that $\widetilde{D}_{\pi_h^a, h'} = \{s_{h',i}\}_{i=1}^{N}$ are the states at level $h'$ on the $N$ trajectories collected using $\pi_h^a$ (cf. Line 4 of Algorithm 3). Again by Lemma A.2, with high probability we have

$$x^\top \left( \mathbb{E}_{s_{\pi_{h'}} \sim \mathcal{D}_{h'}^{\pi_h^a}} \left[ \phi(s_{\pi_{h'}}) \phi(s_{\pi_{h'}})^\top \right] \right) x \geq \frac{\epsilon_t}{2}.$$

Thus, if we use $\mathcal{M}$ to denote

$$\left\{ \mathbb{E}_{s_{\pi_{h'}} \sim \mathcal{D}_{h'}^{\pi_{h'}}} \left[ \phi(s_{\pi_{h'}}) \phi(s_{\pi_{h'}})^\top \right] \right\}_{\pi_{h'} \in \Pi_{h'}},$$

and $M_t$ to denote

$$\mathbb{E}_{s_{\pi_{h'}} \sim \mathcal{D}_{h'}^{\pi_h^a}} \left[ \phi(s_{\pi_{h'}}) \phi(s_{\pi_{h'}})^\top \right],$$

then this is exactly the process described in Lemma A.5, and thus the upper bound on $|\Pi_{h'}|$ follows. $\square$