[Reviews · NeurIPS 2019]

Reviewer 1



***Post rebuttal*** I read the other reviews and the author feedback. I appreciate the effort of the authors in addressing my issues. The authors managed to provide significant examples in which the performance gap is finite. Still, I believe that the assumption might be restrictive and that the knowledge of the performance gap (or some lower bound) is not straightforward. The authors clarified my confusion about Section 5. I am raising the score to 6. I think that the paper provides nice contributions, although the resulting algorithm DMQ is quite complex and the authors did not completely succeed in explaining the general idea. Moreover, I have some doubts on the assumptions. I made a high-level check of the math and it seems ok. Overall, the paper is written in good English and reads well, although the organization of the contents can be improved. Here are my detailed comments. ***Major*** - Definition 3.1: I am fine with this definition when considering finite state spaces, but it seems to me that the minimum over the state space cannot be so easily used when passing to infinite state spaces. Indeed, it might be the case that such minimum does not exist. In this case, it would be more correct to use the infimum. However, the infimum could be zero and, consequently, the bound of Theorem 6.1 would become vacuous. Are the authors assuming that the gap exists and it is strictly greater than zero? It seems that this is the case as stated in Section 7 (Regret Bound paragraph). If so, I suggest to clearly state the assumption sooner. How much is this assumption restrictive? Moreover, the $\gamma$ is used to define the hyperparameter $\epsilon_t$, which is used by the algorithm. Since $\gamma$ depends on the optimal value function itself, how can $\gamma$ be computed in practice? Are the authors assuming that $\gamma$ is known in advance? - Section 5: the proposed algorithm turned out to be quite elaborate (it is split into three pseudocodes Algorithm 1-3). While Section 5 explains step-by-step the functioning of the algorithm, I found quite hard to understand why certain choices were made. I think that the main problem with this section is that it lacks an overview of the general idea of the algorithm. For instance, it is not immediately clear the motivation behind the definition of the policy at Equation (1) or the actual role of the oracles in the algorithm. ***Medium*** - The abstract seems a bit too long, and dives into many details, like the explanation of the oracles. - Section 1: the authors report in the Introduction the statement, although informal, of Theorem 1.1, which is the main contribution of the paper. I think that reporting it in the introduction is premature, I suggest to describe the meaning of the theorem without the statement. - Oracle 4.1: the regularizer $\Lambda(f_1,f_2)$ is used in the definition but explained only later. I suggest anticipating the definition of $\Lambda$. ***Minor*** - line 146: "By Q-learning, we mean ..." this sentence is not very clear - line 196-197: "does not rely any label information" this sentence is not very clear ***Typos*** - line 97: algorithm. which -> algorithm which - line 216: choose -> chooses - line 216: use -> uses - line 238: in the definition of $M_1$ and $M_2$, the subscripts of $D$ ($1$ and $2$) should appear inside the absolute value - line 272: Note we -> Note that we

Reviewer 2



After rebuttal: I read the author response and other reviews. I'll keep my assessment. --------------------------- This paper studies the Q-learning problem with linear function approximation and proposes the first provably efficient algorithm DMQ for general stochastic setting. Melo and Ribeiro [2007] and Zou et al. [2019] are most related work but assumes the exploration policy is fixed. This paper also introduces a DESC oracle, which examines whether all functions perform well on both distribution D1 and D2. In the DMQ algorithm, the learned predictor will be near optimal when the DESC oracle returns False. In addition, DESC oracle will only return true at most polynomial number of times thus avoiding exponential complexity. The DESC oracle is novel and has a nice property. Overall, this paper is clearly written and theoretically sound. Some comments: The proposed DESC oracle has an interesting property to me. By definition, it tests whether all functions in the function class work well under both distribution D1 and D2. In the DMQ algorithm, it will guarantee the learned \hat{\theta} is close to the true \theta. One limitation of this work is the assumption that the true Q-function is exactly linear, which will not hold in general. Is it possible to extend to the approximate case? Will the algorithm also return near optimal policy when the best linear approximation is close to the true Q-function? The sample complexity is polynomial in terms the 1/\gamma, where \gamma is the suboptimality gap. If few state-action pairs have 0 or quite small gap, but most state-action pairs have large gaps, can the sample complexity be improved? In Algorithm 2, the agent needs to collect on-the-go reward y, which has higher computation complexity than collecting reward r since the agent need to roll out the timestep H. The proof of Theorem 6.1 is a little unclear to me. In line 401 and 402, why label ys in D_h^a have bias bounded by \epsilon? Is it because the result hold for h'=h+1,...,H so that the induction can be applied? The \gamma notation denotes the suboptimality gap in this paper. Although it will not lead to confusion, I think it would be better to change to another notation since \gamma is usually reserved for the discount factor.

Reviewer 3



Overall the paper is clear and well presented. The intuition of the algorithm is also explained. I only have minor comments on the results. 1. Can we make Theorem 6.1 in a more general setting, maybe under some assumption on the error checking oracle or has the number of this oracle been called in the bound? 2. As pointed out by the authors, the proposed algorithm is not optimal. This can be seen from a particular setting: stochastic bandits. It may also help the readers to better understand the algorithm by discussing its behaviour on a stochastic bandits setting. In particular, in order to identify the best arm, Algorithm 2 line 6-8 is not optimal. Have the authors tried to improve this part? 3. Can the results on Lemma A.4 be generalized to other setting rather than linear approximation? ============== After rebuttal: I have read the rebuttal and other reviews. My score remains the same.

[Author Response · NeurIPS 2019]

We thank all reviewers for appreciating our technical contributions: a novel algorithm and a sound theoretical analysis.
Please find our response to each reviewer below.

**To Reviewer #1:**

-*Definition 3.1*: We will state in Definition 3.1 explicitly that we assume $\gamma > 0$ to avoid confusion. Thanks for pointing
out. We will also state that we assume $\gamma > 0$ again before Theorem 6.1 to emphasize this assumption.

-*Assuming $\gamma > 0$ is restrictive:* Assuming $\gamma$ strictly positive is not a restrictive assumption for the *finite action setting*
considered in this paper. First, in the contextual linear bandit literature, this assumption is widely discussed. See, e.g.,
[1,2]. The notion, context, in the bandit literature is essentially $\phi(s)$ in our paper and the number of contexts can also
be infinite. Second, there are many natural environments in RL which satisfy this assumption. For example, in many
environments, states can be classified as good states and bad states. In these environments, an agent can obtain a reward
only if it is in a good state. There are also two kinds of actions: good actions and bad actions. If the agent is in a good
state and chooses a good action, the agent will transit to a good state. If the agent chooses a bad action, the agent will
transit to a bad state. If the agent is in a bad state, whatever action the agent chooses, the agent will transit to a bad state.
Note that for this kind of environments, there is a strictly positive gap between good actions and bad actions when the
agent is in good states and there is no difference between good actions and bad actions when the agent is in bad states.
In this case, $\gamma$ is strictly positive, since by Definition 3.1, we take the minimum over all state-action pairs with *strictly*
positive gap. These environments are natural generalizations of the combination lock environment [3]. Some Atari
games, e.g. Freeway, have a similar flavor as these environments.

-*What if $\gamma$ is unknown:* While our algorithm requires $\gamma$ as an input for deciding hyper-parameters, it is easy to see from
the proof that as long as we choose some $\gamma' \leq \gamma$ and choose hyper-parameters according to $\gamma'$, the sample complexity
bound still holds. Therefore, we can just do a grid search over $[0,1]$ to find such $\gamma'$. We will add more discussion.

-*Section 5:* We will put an overview of our algorithm at the beginning of Section 5. The design of our algorithm is
actually fairly natural as we explain below. Since we know $Q^*$ belongs to a function class, the natural idea is to use
regression to recover it (Line 9 of Algorithm 2), for which we need unbiased samples. To obtain unbiased samples for
level $h$, we need to make sure that we execute the optimal policy for level $h' = h + 1, \ldots, H$ (this accounts for our
definition in Eq. 1). Therefore, in Line 1-2 of Algorithm 1, we learn $Q^*$ from the last level to the first level (learning
at the last level is equivalent to the stochastic bandit problem, where we have unbiased samples). The tricky part is
that the $Q$-function we learned, say at level $h' = h + 1, \ldots, H$, is only guaranteed to be optimal with respect to the
roll-in policy we used to gather samples. When we learn $Q^*$ at level $h$, the roll-in policy for level $h' = h + 1, \ldots, H$
can change. To address this problem, we need to check that the learned $Q$-function is still optimal for the new roll-in
policy (Algorithm 3 and the DSEC oracle). We have explained the intuition of the DSEC oracle in Section 1.

-*Abstract / introduction:* We follow one of the common writing styles that first states our precise result and gives a
description of our main techniques in the abstract, and then puts an informal main theorem and a technical overview in
the introduction. We are happy to adjust our writing style.

-*Regularizer in Oracle 4.1 / minor concerns / typos*: We will fix them accordingly. Thanks for pointing out.

**To Reviewer #2**:

-*Q-function is not exactly linear:* This is a great question. Our algorithm can still be applied if the best linear predictor
has approximation error smaller than $\frac{\gamma}{\text{poly}(H,d)}$. The proof is essentially the same. We will add more discussion.

-*Most state-action pairs have a large gap:* This is an interesting question. We do believe the sample complexity can be
improved with proper quantification on "most state-action pairs". We will discuss this in the final version.

-*Need to collect on-the-go reward*: We agree this incurs higher computation complexity. We will list improving the
computation complexity as an open problem.

-*Proof of Theorem 6.1:* In Line 401 and 402 we use the induction hypothesis of later levels. We will add more details.

-*Notation $\gamma$:* Thanks for pointing out. We will change the notation to $\triangle$.

**To Reviewer #3**:

-*More general setting / generalization of Lemma 4:* Thanks for the question! We have discussed going beyond linear
function approximation in Section 7. A natural extension of Lemma 4 is to study the Fisher information matrix, though
this generalization is not trivial. Once we know the number of calls to the DSEC oracle is bounded (via a generalization
of Lemma 4), we can then reuse the current proof framework.

-*Optimality / connections with stochastic bandits:* We will discuss more about connections with stochastic bandits.
Thanks for the suggestion. While Line 6-8 is not optimal for the stochastic bandit case, it not obvious how to improve
that part since Algorithm 3 also depends on the data collection policy. We will add more discussion.

[1] Yasin Abbasi-Yadkori, Dávid Pál, and Csaba Szepesvári. Improved algorithms for linear stochastic bandits.
[2] Varsha Dani, Thomas P Hayes, and Sham M Kakade. Stochastic linear optimization under bandit feedback.
[3] Sham M Kakade. On the sample complexity of reinforcement learning.


[Meta-Review · NeurIPS 2019]

The paper proposes an adaptation of the classical Q-learning algorithm with linear function approximation that enjoys polynomial sample complexity. All reviewers feel the paper contains interesting contribution to the RL literature that should appear in this conference, and I therefore recommend acceptance.